# `stable-pretraining-v1`:
# Foundation Model Research Made Simple

**Randall Balestriero**[1][*], **Hugues Van Assel**[2], **Sami BuGhanem**[1], **Lucas Maes**[3]
[1]Brown University, [2]Genentech, [3]Mila & Université de Montréal

## Abstract

Foundation models and self-supervised learning (SSL) have become central to modern AI, yet research in this area remains hindered by complex codebases, redundant re-implementations, and the heavy engineering burden of scaling experiments. We present `stable-pretraining`, a modular, extensible, and performance-optimized library built on top of PyTorch, Lightning, Hugging Face, and TorchMetrics. Unlike prior toolkits focused narrowly on reproducing state-of-the-art results, `stable-pretraining` is designed for flexibility and iteration speed: it unifies essential SSL utilities—including probes, collapse detection metrics, augmentation pipelines, and extensible evaluation routines—within a coherent and reliable framework. A central design principle is *logging everything*, enabling fine-grained visibility into training dynamics that makes debugging, monitoring, and reproducibility seamless. We validate the library by demonstrating its ability to generate new research insights with minimal overhead, including depthwise representation probing and the analysis of CLIP degradation under synthetic data finetuning. By lowering barriers to entry while remaining scalable to large experiments, `stable-pretraining` aims to accelerate discovery and expand the possibilities of foundation model research. The source code is available at https://github.com/rbalestr-lab/stable-pretraining.

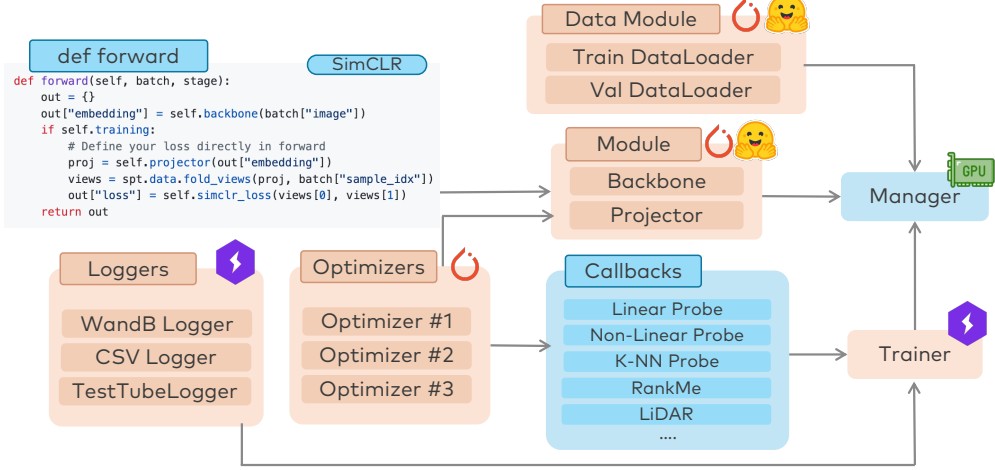

Figure 1: Overview of `stable-pretraining`.

[*]Correspondence to `rbalestr@brown.edu`

# 1 Introduction

Foundation models have transformed artificial intelligence in the past decade, powering breakthroughs across vision, language, and multimodal learning. Yet, despite this progress, research on foundation models remains uniquely challenging. Unlike conventional supervised learning, it requires large-scale datasets, multi-GPU training setups, and intricate monitoring of training dynamics. Researchers must navigate debugging difficulties, collapse detection, careful hyperparameter tuning, and complex evaluation protocols [2]—none of which are readily supported in mainstream frameworks like PyTorch [12], Lightning[6], or Hugging Face[18, 10]. As a result, even simple experiments often demand starting from massive, monolithic codebases such as DINOv2 [11] or MAE [9]. These repositories are difficult to extend, tightly coupled to specific engineering choices, and slow to prototype with—creating a bottleneck for innovation. Compounding the problem, many research groups repeatedly re-implement the same essential components: data augmentation pipelines, training loops, probes, loss functions, or evaluation metrics. This redundancy is not only inefficient but also increases the likelihood of bugs, inconsistencies, and incomparable evaluation results across the community. The consequence is *a research ecosystem constrained to incremental improvements, with limited room for rapid exploration of new ideas*.

Several prior libraries have attempted to address these challenges, such as VISSL [8], solo-learn [4], or lightly [15]. However, these toolkits share important limitations, e.g., they are static by design, focusing on reproducing established methods rather than supporting new research exploration. Moreover, VISSL and solo-learn are no longer actively maintained, with their last commits dating back to 2022 and 2023, respectively. Lightly, on the other hand, separates SSL functionality from training utilities, many of which are only accessible through paid membership. Lastly, none of these frameworks treat monitoring and debugging as first-class concerns, leaving researchers to repeatedly engineer their own probes, evaluation pipelines, or collapse detection metrics. As a result, *existing solutions only partially reduce the engineering burden and do not fully support the rapid, exploratory workflows needed for foundation model research*.

To address these challenges, we present `stable-pretraining`, a library purpose-built for rapid and scalable foundation model research. Built on top of PyTorch, Lightning, Hugging Face, and TorchMetrics [5], it combines the reliability of widely adopted frameworks with specificities required for foundation model training, typically absent elsewhere. Unlike prior toolkits focused narrowly on reproducing state-of-the-art results, `stable-pretraining` is designed for flexibility and iteration speed. Its modular framework consolidates critical SSL components—including probes (linear, non-linear, $k$-NN), collapse detection metrics (RankMe [7], LiDAR [16]), and extensible evaluation utilities—into a unified, performance-optimized system. At its core, `stable-pretraining` logs every aspect of training and evaluation, providing fine-grained monitoring and transparent feedback that facilitates debugging, reproducibility, and deeper insights from training dynamics. Our goal is to expand what is possible in foundation model research: to accelerate discovery, foster reproducibility, and empower the community to explore beyond today's incremental progress.

Table 1: Linear probe top-1 accuracy across multiple datasets.

| Method | Arch. | DTD | aircraft | cars | cifar10 | cifar100 | flowers102 | food101 | galaxy10 | pets | avg. |
|---|---|---|---|---|---|---|---|---|---|---|---|
| I-JEPA [1] | ViT-H | 73.62 | 56.45 | 58.93 | 97.77 | 86.93 | 85.76 | 81.06 | 62.93 | 92.94 | 77.37 |
| DINO [3] | ViT-S | 77.29 | 72.92 | 75.86 | 97.12 | 85.27 | 95.13 | 84.81 | **68.91** | 95.00 | 83.59 |
| DINOv2 [11] | ViT-S | **80.43** | **80.56** | **84.21** | **97.75** | **88.04** | **99.56** | **90.52** | 67.60 | **95.67** | **87.15** |

# 2 `stable-pretraining`: An Overview

`stable-pretraining`'s focus is to alleviate the tedious process of assembling a foundation model research codebase. We argue that the lack of such library poses an important limitation in current research as the barrier to entry has become insurmountable. With our solution, the time from research idea to first sign of success of failure is drastically reduced. In the following sections, we first outline the design choices behind `stable-pretraining`. We then highlight our research utilities by presenting two simple yet previously unverified experimental insights in self-supervised learning.

## 2.1 Structure

Figure 1 provides an overview of `stable-pretraining`. Our design philosophy is simple: reuse what the community already trusts, and build only what is missing to perform efficient research. Components shown in blue represent modules we specifically developed, while those in orange are borrowed and adapted from proven third-party libraries such as Lightning, Hugging Face, and PyTorch. At the center of the pipeline is the `Manager`, a lightweight controller that works in tandem with Lightning's `Trainer` to coordinate the entire training process. The Manager abstracts away many tedious engineering details—such as automatic checkpoint handling in cluster environments, consistent logging, and monitoring (optional)—so that researchers can focus on experimentation rather than infrastructure.

**Manager and logging-everything.** The `Manager` works synergistically with Lightning's `Trainer` to orchestrate the entire training pipeline, handling model execution, checkpointing, and environment-specific details such as automated reloads on clusters. At the same time, it embodies our *log-everything* ethos as a first-class concern: every component of the pipeline is logged in a fine-grained and structured manner. This design turns monitoring, reproducibility, and debugging into routine features rather than burdens, aligning the library's ergonomics with the pace and reliability needs of rapid foundation model research.

**Dictionary-first design.** Everything in `stable-pretraining` speaks dictionaries. Datasets emit dictionary-shaped batches; modules consume and produce dictionaries; callbacks read/write named fields. Common keys include `image`, `label`, `embedding`, `loss`. This uniform interface removes glue code, keeps components swappable, and makes pipelines easy to extend.

**Data and module composition.** The `DataModule` encapsulates training and validation dataloaders (e.g., from Hugging Face datasets or custom sources). The `Module` bundles any number of PyTorch components (such as backbones, projectors, classifiers, or losses) and orchestrates their interaction through a user-defined `forward(self, batch, stage)`. Unlike PyTorch Lightning, where one must implement separate `training_step`, `validation_step`, and related methods, this framework consolidates all computation in the `forward` function. The `forward` not only produces embeddings, predictions, or other intermediate representations, but can also compute losses directly when invoked during training. The return value is a dictionary that may contain arbitrary keys (e.g., "embedding", "prediction") for monitoring and analysis, with the special convention that a ``loss`` key—if present—will be used automatically for optimization. This design keeps training logic explicit and flexible while avoiding boilerplate, and it ensures that outputs, metrics, and losses are unified in a single, stage-aware interface.

**Callbacks.** A major convenience of our library is its set of plug-and-play callbacks for monitoring and evaluation: linear and non-linear (attentive) probes, $k$-NN probes, and collapse detection metrics (RankMe, LiDAR), among others. The callback engine is backed by an intelligent, shared-memory queue: when multiple callbacks consume the same tensors (e.g., embeddings), computations are deduplicated and memory is reused. Our callbacks deliver (i) real-time feedback on representation quality, (ii) early detection of collapse, and (iii) multi-metric views that turn debugging into insight—with minimal overhead. Importantly, all callbacks are implemented as native Lightning callbacks, ensuring full compatibility: researchers can freely mix and match our probes and monitors with any standard or custom Lightning callback in a single training loop.

## 2.2 Accelerating Research

Beyond faithfully reproducing existing approaches, `stable-pretraining` is designed to accelerate the process of exploring new ideas. Its modularity and plug-and-play utilities enable experiments that would otherwise require considerable and repetitive effort to be carried out with minimal setup. We illustrate this through two case studies. As a sanity check, we also report the linear probe accuracy over a wide range of datasets for different methods in table 1.

**Depth-wise representation probing.** Analyzing intermediate representations in large models typically demands intrusive modifications to training code and custom evaluation pipelines. With `stable-pretraining`, this becomes trivial: adding a linear probe at arbitrary layers requires only a

few lines of configuration. As a demonstration, we probe ImageNet-100 representations at multiple depths across several state-of-the-art vision SSL models. Results (Figure 2) confirm the expected trend that later layers yield stronger performance, while also revealing that MetaCLIP [19] excels at earlier and intermediate layers, whereas DINOv2-3 [11, 14] dominates at the final layer. This experiment, often prohibitively cumbersome, is reduced to a straightforward plug-and-play setup.

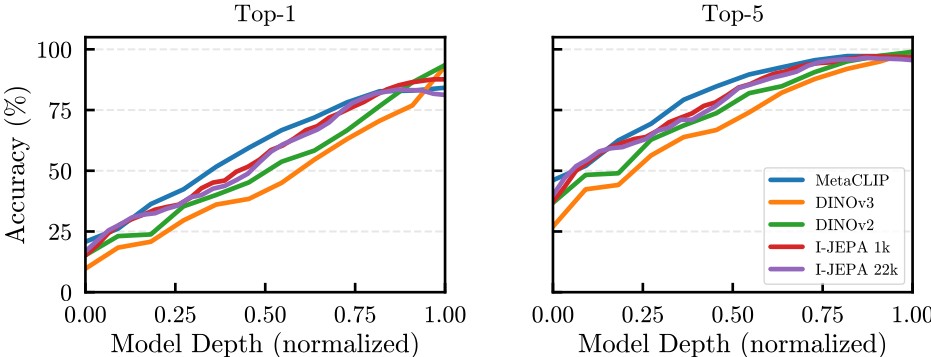

Figure 2: Depth-wise representation probing (ImageNet-100). We report the top-1 and top-5 validation accuracies of linear probes from different layers of SOTA vision self-supervised learning methods after 100 epochs. MetaCLIP outperforms other approaches on beginning and intermediate layers, while DINOv2-3 outperforms on the last layer.

**CLIP degradation under synthetic data fine-tuning.** We further showcase how `stable-pretraining` facilitates rapid exploration of new research questions. Starting from a frozen CLIP ViT-B/32 [13] checkpoint, we continue pretraining for 8 epochs on a synthetic image dataset DiffusionDB-2M [17], monitoring zero-shot transfer throughout. As shown in Table 2, performance degrades sharply: Top-1 accuracy on ImageNet-100 drops by 19% after just a single epoch, with continued training yielding no recovery. This highlights how quickly synthetic data can harm representation quality in self-supervised learning—a result that can be obtained with minimal overhead using our framework.

Table 2: CLIP (OpenAI clip-vit-base-patch32 d) model accuracies on ImageNet-100 validation set before and after finetuning on DiffusionDB2M. Synthetic data fine-tuning seems to degrade the quality of learned SSL representations.

| Metric | No Finetuning | Epoch 1 | Epoch 2 | Epoch 3 | Epoch 4 | Epoch 5 | Epoch 6 | Epoch 7 | Epoch 8 |
|--------|--------------|---------|---------|---------|---------|---------|---------|---------|---------|
| Top-1  | **77.7**     | 59.0    | 50.3    | 48.2    | 50.1    | 49.7    | 49.9    | 50.2    | 50.4    |
| Top-5  | **94.6**     | 87.4    | 81.9    | 79.8    | 81.5    | 80.4    | 80.2    | 80.8    | 81.3    |
| Top-10 | **97.3**     | 93.7    | 89.8    | 88.8    | 90.0    | 89.3    | 88.7    | 88.6    | 89.3    |

# 3 Conclusion

We introduced `stable-pretraining`, an open-source library designed to accelerate and simplify research on foundation models and self-supervised learning. Built on top of PyTorch, Lightning, Hugging Face, and TorchMetrics, it ensures stability and extensibility while avoiding redundant engineering. Unlike prior efforts focused on reproducing existing methods, `stable-pretraining` emphasizes flexibility, iteration speed, and modularity by consolidating essential SSL utilities—such as probes, collapse detection, and extensible evaluation pipelines—into a unified, performance-optimized framework. A central design principle is *logging everything*, making monitoring and debugging transparent, reproducible, and directly useful for research. We validate that the library not only reproduces state-of-the-art performance but also enables new research insights with minimal effort, lowering the barrier to entry while supporting large-scale experimentation.

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

## A    Code snippets.

```python
import lightning as pl
import torch
import torchmetrics
import torchvision
from torch import nn
from lightning.pytorch.loggers import WandbLogger

import stable_pretraining as spt
from stable_pretraining.data import transforms

# Define augmentations for SimCLR (creates 2 views of each image)
simclr_transform = transforms.MultiViewTransform(
    [
        transforms.Compose(
            transforms.RGB(),
            transforms.RandomResizedCrop((32, 32), scale=(0.2, 1.0)),
            transforms.RandomHorizontalFlip(p=0.5),
            transforms.ColorJitter(brightness=0.4, contrast=0.4, saturation=0.2,
    hue=0.1, p=0.8),
            transforms.RandomGrayscale(p=0.2),
            transforms.ToImage(**spt.data.static.CIFAR10),
        ),
        # Second view with slightly different augmentations
        transforms.Compose(
            transforms.RGB(),
            transforms.RandomResizedCrop((32, 32), scale=(0.08, 1.0)),
```

```
26              transforms.RandomHorizontalFlip(p=0.5),
27              transforms.ColorJitter(brightness=0.4, contrast=0.4, saturation=0.2,
        hue=0.1, p=0.8),
28              transforms.RandomGrayscale(p=0.2),
29              transforms.RandomSolarize(threshold=0.5, p=0.2),
30              transforms.ToImage(**spt.data.static.CIFAR10),
31          ),
32      ]
33  )
34
35  # Load CIFAR-10 and wrap in dictionary format
36  cifar_train = torchvision.datasets.CIFAR10(train=True)
37  cifar_val = torchvision.datasets.CIFAR10(train=False)
38
39  train_dataset = spt.data.FromTorchDataset(
40      cifar_train,
41      names=["image", "label"],  # Convert tuple to dictionary
42      transform=simclr_transform,
43  )
44
45  val_dataset = spt.data.FromTorchDataset(
46      cifar_val,
47      names=["image", "label"],
48      transform=transforms.Compose(
49          transforms.RGB(),
50          transforms.Resize((32, 32)),
51          transforms.ToImage(**spt.data.static.CIFAR10),
52      ),
53  )
54
55  # Create dataloaders with view sampling for contrastive learning
56  train_dataloader = torch.utils.data.DataLoader(
57      dataset=train_dataset,
58      sampler=spt.data.sampler.RepeatedRandomSampler(train_dataset, n_views=2),
59      batch_size=256,
60      num_workers=8,
61      drop_last=True,
62  )
63
64  val_dataloader = torch.utils.data.DataLoader(
65      dataset=val_dataset,
66      batch_size=256,
67      num_workers=10,
68  )
69
70  data = spt.data.DataModule(train=train_dataloader, val=val_dataloader)
71
72  # Define the forward function (replaces training_step in PyTorch Lightning)
73  def forward(self, batch, stage):
74      out = {}
75      out["embedding"] = self.backbone(batch["image"])
76      if self.training:
77          # Project embeddings and compute contrastive loss
78          proj = self.projector(out["embedding"])
79          views = spt.data.fold_views(proj, batch["sample_idx"])
80          out["loss"] = self.simclr_loss(views[0], views[1])
81      return out
82
83  # Build model components
84  backbone = spt.backbone.from_torchvision("resnet18", low_resolution=True)
85  backbone.fc = torch.nn.Identity()  # Remove classification head
86
87  projector = nn.Sequential(
88      nn.Linear(512, 2048),
89      nn.BatchNorm1d(2048),
```

```
90          nn.ReLU(inplace=True),
91          nn.Linear(2048, 2048),
92          nn.BatchNorm1d(2048),
93          nn.ReLU(inplace=True),
94          nn.Linear(2048, 256),
95      )
96
97      # Create the module with all components
98      module = spt.Module(
99          backbone=backbone,
100         projector=projector,
101         forward=forward,
102         simclr_loss=spt.losses.NTXEntLoss(temperature=0.5),
103         optim={
104             "optimizer": {"type": "LARS", "lr": 5, "weight_decay": 1e-6},
105             "scheduler": {"type": "LinearWarmupCosineAnnealing"},
106             "interval": "epoch",
107         },
108     )
109
110     # Add callbacks for monitoring performance during training
111     linear_probe = spt.callbacks.OnlineProbe(
112         name="linear_probe",
113         input="embedding",
114         target="label",
115         probe=torch.nn.Linear(512, 10),
116         loss_fn=torch.nn.CrossEntropyLoss(),
117         metrics={
118             "top1": torchmetrics.classification.MulticlassAccuracy(10),
119             "top5": torchmetrics.classification.MulticlassAccuracy(10, top_k=5),
120         },
121     )
122
123     knn_probe = spt.callbacks.OnlineKNN(
124         name="knn_probe",
125         input="embedding",
126         target="label",
127         queue_length=20000,
128         metrics={"accuracy": torchmetrics.classification.MulticlassAccuracy(10)},
129         input_dim=512,
130         k=10,
131     )
132
133     # Configure training
134     trainer = pl.Trainer(
135         max_epochs=1000,
136         callbacks=[knn_probe, linear_probe],   # Monitor SSL quality in real-time
137         precision="16-mixed",
138         logger=WandbLogger(project="cifar10-simclr"),
139     )
140
141     # Launch training
142     manager = spt.Manager(trainer=trainer, module=module, data=data)
143     manager()
```

Listing 1: SimCLR training on CIFAR-10 with `stable_pretraining` and PyTorch Lightning

