# OpenReview forum: "stable-pretraining: Foundation Model Research Made Simple"
_NeurIPS.cc/2025/Workshop/UniReps — UniReps2025_

### Official Review · Reviewer_6CqP · 2025-09-14
**stable-pretraining: Foundation Model Research Made Simple**

**Confidence:** 4

**Review:**

This paper presents stable-pretraining, an open-source library for self-supervised and foundation model research. It integrates PyTorch, Lightning, Hugging Face, and TorchMetrics into a modular framework. The design emphasizes iteration speed, reproducibility, and flexibility by unifying utilities like probes, collapse detection (RankMe, LiDAR), augmentation pipelines, and extensible evaluation. The paper validates the library with two case studies: (1) depth-wise probing across vision SSL models, and (2) CLIP ViT-B/32 degradation when fine-tuned on synthetic DiffusionDB-2M data.
like the paper’s pragmatism: Figure 1 shows a design that actually mirrors how many of us prototype SSL—Manager steering Lightning’s Trainer, everything passing neat dictionaries (image, embedding, loss), and callbacks doing the heavy lifting. The “log everything” principle directly addresses the real pain points of debugging collapse and probe drift.

The dictionary-first design and single forward(self, batch, stage) interface make pipelines cleaner than Lightning’s multiple *_step methods. Listing 1 also reads very natural: wrap CIFAR-10 in dicts, plug in SimCLR augmentations, attach OnlineKNN + linear probe, and launch with a one-liner Manager(). That’s a strong reproducibility win.

The two case studies are nice proofs of utility. The depth-wise probing result (MetaCLIP better in early/mid layers, DINOv2/3 dominating at the final layer) is something I’ll probably cite. And the CLIP-on-synthetic experiment is a memorable cautionary tale: Top-1 accuracy plunges from 77.7% to 59.0% after a single epoch, never fully recovering even after 8 epochs.

Strengths

Clear writing and strong motivation (why existing VISSL/solo-learn/lightly are insufficient).
Modular and extensible: Manager, callbacks, and dict interface feel ergonomic.
“Log everything” principle directly supports reproducibility and monitoring.
Case studies demonstrate both expected trends and surprising insights.
Code examples (SimCLR+CIFAR-10) are immediately usable.

Weaknesses / Questions

Shared-memory queue: How large is the queue for OnlineKNN + linear probe, and what is the VRAM/runtime overhead vs. running them separately? Any ablation?
Scaling evidence: The claim is that the library scales to large models, but experiments are at ImageNet-100 and CLIP-B/32 scale. Can the authors show throughput on something larger (e.g., ViT-L, multi-node) or at least a small runtime table vs. VISSL/solo-learn/lightly?
CLIP fine-tuning degradation: Is the sharp drop due mainly to distribution shift in DiffusionDB prompts or the optimizer schedule? A control with reduced LR or partial synthetic mixing would clarify.

Minor nitpicks:
Line 63: “sign of success of failure” → should be success or failure.

Pros
Strong engineering quality, modular design, and extensibility.
Directly addresses reproducibility/debugging pain points.
Open-source and easy to adopt.
Demonstrates real research insights with minimal engineering.

Cons
Algorithmic novelty is limited.
Large-scale evidence is missing.
No runtime/UX comparison with prior libraries.
Learning curve for beginners in Lightning/Hugging Face not discussed.

**Score:**

3

**Topic Fit:**

2

---

### Official Review · Reviewer_nC5Q · 2025-09-15
**Review on the paper**

**Confidence:** 4

**Review:**

The paper presents stable-pretraining, a valuable and well-designed library that effectively addresses the significant engineering overhead in foundation model research.
The work's primary strength is its practical, modular design built upon trusted frameworks like PyTorch and Lightning, which simplifies a complex workflow. Its focus on research iteration through unified utilities like online probes and collapse-detection callbacks is a significant contribution that lowers the barrier to entry and can accelerate discovery in the field. The case studies effectively demonstrate the library's utility for generating novel insights with minimal effort.
While the contribution is strong, the paper could be improved by briefly addressing two points. First, considering that similar libraries have become inactive, a short discussion on the strategy for long-term maintenance would reassure potential users. Second, clarifying why a vision-based SimCLR example was chosen, when the term "pretraining" is often associated with LLMs, would better frame the library's general-purpose applicability.
These are minor points for consideration. The paper presents a timely and useful tool that represents a clear contribution to the research community.

**Score:**

4

**Topic Fit:**

2

---

### Official Review · Reviewer_QWZX · 2025-09-15
**interesting**

**Confidence:** 3

**Review:**

Summary of the work

The paper introduces stable-pretraining, a modular PyTorch/Lightning/HuggingFace/TorchMetrics based library that aims to lower the engineering overhead of self‑supervised/foundation‑model research. Core ideas include dictionary‑first interface for batches, modules, and callbacks; a unified `forward(self, batch, stage)` entry point; and callbacks for online probes (linear, non‑linear, k‑NN) and collapse‑detection metrics (RankMe, LiDAR). The EA validates the library via two compact case studies: (1) depth‑wise linear probing across SOTA vision SSL models on ImageNet‑100, and (2) zero‑shot degradation of CLIP after finetuning on synthetic images from DiffusionDB‑2M .

Overall assessment

- **Clarity:** Strong. The high‑level design and the 'dictionary‑first + callbacks' message are easy to grasp; the SimCLR listing makes the API concrete. Some experimental details are unclear (see below), but that is acceptable for an extended abstract and fixable with small edits.
- **Novelty:** Moderate‑to‑good. As a *researcher‑oriented* library rather than a reproduction toolkit, the Manager + logging focus and the stage‑aware `forward` are thoughtful design choices. The systems contribution is incremental relative to VISSL/solo‑learn/lightly, but the *research‑workflow* emphasis feels new
- **Potential to inspire discussion:** High. The case studies (depth‑wise probing; synthetic‑data finetuning harms CLIP zero‑shot) open several “why” questions that the community may want to debate and extend.

Strengths

1. The Manager abstracts away cluster/resume/logging concerns and coordinates with Lightning’s `Trainer`
2. The dictionary‑first design makes callbacks composable
3. Built‑in monitoring & probes. Real‑time linear/non‑linear/k‑NN probes and RankMe/LiDAR collapse monitors, with shared‑tensor deduplication, are valuable QoL features.
4. Concrete, reproducible‑looking examples. depth‑wise probes and CLIP on DiffusionDB‑2M are convincing; Listing 1 shows that reproducing SimCLR on CIFAR‑10 truly is really just a few dozen lines.

Concerns
1. Scope: "foundation models" vs. current coverage (mostly vision).
    The abstract reads broadly (foundation models, multimodal), but all concrete results are vision (ImageNet‑100; CLIP ViT‑B/32). Please state the current scope explicitly (e.g., "Vision SSL now; easy hooks for text/multimodal via HF Transformers") and list at least one text‑or multimodal example in the repo (even a tiny masked‑LM probe) to align claims with evidence.
2. Synthetic‑data finetuning details are ambiguous.
    Table 1 shows a 19 point drop in ImageNet‑100 top 1 after 1 epoch of synthetic finetuning, but the text says "starting from a frozen CLIP… we continue pretraining". Clarify:

   - Which towers are updated (vision only? text frozen?) and what objective is used (image‑only SSL vs. CLIP contrastive with prompts from DiffusionDB)?
     Are zero‑shot accuracies computed with the standard CLIP prompt set?
3. Experimental specifics missing for Fig. 2 (depth‑wise probes). Please specify: (i) how normalized depth is defined across architectures; (ii) which checkpoints/datasets for MetaCLIP, DINOv2/3, I‑JEPA ; (iii) linear‑probe protocol (features frozen? epochs 100 are for pretraining or probe?).
4. Positioning vs. prior toolkits would benefit from a comparison.
    The intro argues that VISSL/solo‑learn are stale and lightly paywalls some utilities (p. 2). Consider adding a 1‑row per toolkitcomparison (features, maintenance status, probes/metrics, logging, license)

**Recommendation: Weak Accept.** A thoughtfully designed, researcher‑centric toolkit with a clear message. The two case studies are interesting and likely to create discussion. With a brief scope clarification and a few tweaks, I recommend to **accept** this as an extended abstract.

**Score:**

3

**Topic Fit:**

2